# GUI-PRA: Process Reward Agent for GUI Tasks

## Abstract

Graphical User Interface (GUI) Agents powered by Multimodal Large Language Models (MLLMs) show significant potential for automating tasks. However, they often struggle with long-horizon tasks, leading to frequent failures. Process Reward Models (PRMs) are a promising solution, as they can guide these agents with crucial process signals during inference. Nevertheless, their application to the GUI domain presents unique challenges. When processing dense artificial inputs with long history data, PRMs suffer from a "lost in the middle" phenomenon, where the overwhelming historical context compromises the evaluation of the current step. Furthermore, standard PRMs lacks GUI changing awareness, providing static evaluations that are disconnected from the dynamic consequences of actions, a critical mismatch with the inherently dynamic nature of GUI tasks. In response to these challenges, we introduce **GUI-PRA** (Process Reward Agent for GUI Tasks), a judge agent designed to better provide process reward than standard PRM by intelligently processing historical context and actively perceiving UI state changes. Specifically, to directly combat the "lost in the middle" phenomenon, we introduce a dynamic memory mechanism consisting of two core components: a Relevance-based Retrieval Module to actively fetch pertinent information from long histories and a Progressive Summarization Module to dynamically condense growing interaction data, ensuring the model focuses on relevant context. Moreover, to address the lack of UI changing awareness, we introduce an Aadaptive UI Perception mechanism. This mechanism enables the agent to reason about UI state changes and dynamically select the most appropriate tool to gather grounded visual evidence, ensuring its evaluation is always informed by the current UI context. To validate the practical utility of our approach, we conduct experiments on two online benchmarks for GUI task. Our best results demonstrate an average success rate improvement of 14.53% across the two benchmarks, a significant outperformance of the 8.56% gain achieved by the standard PRM baseline.

## 1 Introduction

Graphical User Interface (GUI) Agents (Hu et al., 2025a; Li et al., 2024b), powered by the rapid development of Multimodal Large Language Models (MLLMs) (Zhang et al., 2025; Li et al., 2024a; Zheng et al., 2024), are emerging as a powerful paradigm for automating complex digital tasks. By leveraging the advanced reasoning, perception, and action capabilities inherent to MLLMs, these agents can interpret and interact with graphical environments at a level approaching human proficiency. Consequently, developing capable GUI Agents is now considered a promising pathway toward more general and autonomous artificial intelligence.

While numerous efforts have sought to improve GUI automation accuracy, many have centered on training-based approaches, such as Supervised Fine-Tuning (SFT) (Gunel et al., 2021; Prottasha et al., 2022) and Reinforcement Learning (RL) Kaelbling et al. (1996); Li (2018). However, these methods often demand extensive, high-quality data and substantial computational resources. This raises a critical question: how can the performance of a pre-existing GUI Agent be enhanced at inference time, without the need for further training? This has led to a growing interest in training-free techniques that can improve agent capabilities on-the-fly. For instance, many studies leverage the ReAct paradigm (Yao et al., 2023), which enables agents to create and adjust plans by cyclically reasoning about their actions and observations. Another popular approach involves decomposing complex GUI tasks and employing a multi-agent Ye et al. (2025); Zhu et al. (2025b) system, where specialized agents for planning and execution collaborate to accomplish the goal. Distinct from

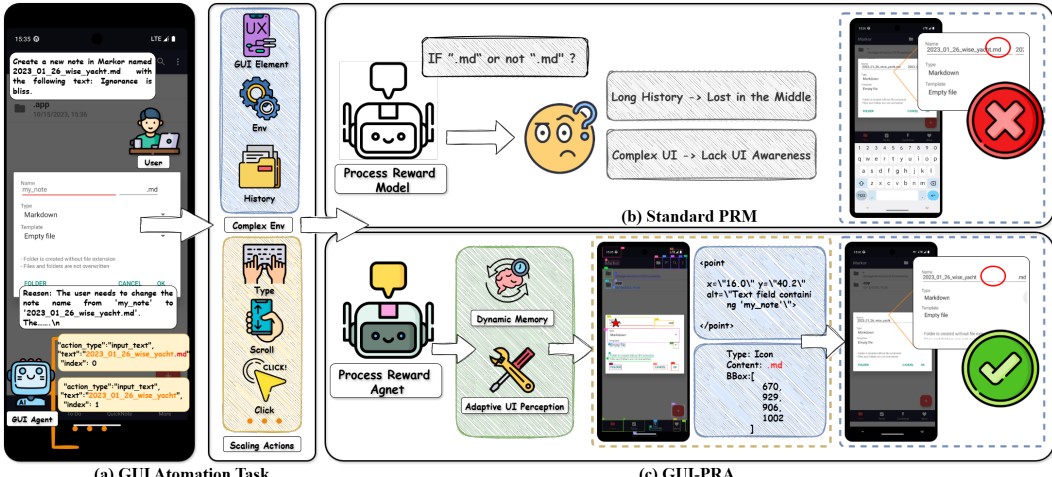

Figure 1: An overview of the GUI-PRA compared to a standard Process Reward Model (PRM). A standard PRM fails a GUI task due to context loss and lack of UI awareness. Our GUI-PRA overcomes these limitations with its Dynamic Memory and UI Tool Routing mechanisms to ensure success.

these methods that enhance an agent's intrinsic reasoning or collaborative structure, another powerful strategy is to introduce an external supervisor.

In general-purpose domains, Process Reward Models (PRMs) Gandhi et al. (2025); Wanyan et al. (2025) have emerged as a highly effective training-free, test-time technique. By providing supervision on an agent's intermediate steps, PRMs can guide it towards a more optimal trajectory from a set of potential action sequences. However, the adaptation of standard PRMs to the distinct challenges of the GUI domain remains under-explored, and they exhibit critical limitations in this context. A primary issue is their struggle with long-context tasks, leading to a "lost in the middle" phenomenon where the model's ability to evaluate the current action is compromised by an overwhelming amount of historical data. Furthermore, standard PRMs lack UI changing awareness. They provide static evaluations based on a textual history, creating a fundamental mismatch with the dynamic nature of GUI tasks where a single action can substantially alter the visual environment. This poses a significant challenge, as the PRM's reward signal becomes disconnected from the visual reality of the task.

In light of these shortcomings, we introduce **GUI-PRA** (Process Reward Agent for GUI Tasks), a training-free framework that transforms a standard PRM into a GUI-domain-specific supervisor. As illustrated in Figure 1, our design achieves this through two core technical contributions, each tailored to address a specific limitation of standard PRMs. First, to address the "lost in the middle" phenomenon, we design a **Dynamic Memory mechanism**. This mechanism intelligently processes the dense historical trajectory by employing two components: a Relevance-based Retrieval Module to filter and retain the most recent and pertinent steps, and a Progressive Summarization Module to condense the long-term interaction history into a concise narrative. This ensures that the agent's evaluation is always based on the most salient historical context. Second, to overcome the PRM's lack of UI changing awareness, we introduce an **Adaptive UI Perception mechanism**. Instead of passively evaluating based on text, this mechanism enables GUI-PRA to actively reason about the UI state. It autonomously selects from a suite of complementary tools—such as OmniParserV2 (Lu et al., 2024) for global UI analysis and Point for fine-grained, localized element grounding—to gather grounded visual evidence. This ensures that its supervision is always informed by the current visual reality of the task. Collectively, these components transform a standard PRM into a dynamic and perceptive agent for GUI tasks.

To validate the effectiveness of our GUI-PRA framework, we conduct a comprehensive evaluation using models from two prominent series, Qwen2.5-VL (Bai et al., 2025) and InternVL (Zhu et al., 2025a; Wang et al., 2025), serving as both the base GUI Agent and the PRM. Our experiments are performed on two online GUI benchmarks: AndroidWorld (Rawles et al., 2025) and Mobile-

MiniWoB++ (Liu et al., 2018). The experimental results demonstrate the clear superiority of our approach. Specifically, GUI-PRA boosts the average success rate of Qwen2.5-VL-7B-Instruct by 14.53% across both benchmarks, significantly surpassing the 8.56% improvement obtained with a standard PRM baseline. Therefore, our core contributions are as follows:

- We propose GUI-PRA, a novel agent that surpass standard PRMs for GUI tasks. This agent is adept at handling dynamic, multi-step tasks, providing better process reward.
- We design two core mechanisms to address the key limitations of standard PRMs: a **Dynamic Memory** mechanism to mitigate the "lost in the middle" problem, and an **Adaptive UI Perception** mechanism to provide awareness of UI state changes.
- We provide extensive empirical validation on AndroidWorld and MobileMiniWob++. Our best results show an average success rate improvement of 14.53%, significantly surpassing the 8.56% gain from a standard PRM baseline.

## 2 PRELIMINARY

### 2.1 GUI TASK AUTOMATION

We study GUI task automation: given a natural language goal description $g$ and an initial GUI state represented by its screenshot and GUI elements $(scr_0, e_0)$, the agent must generate a sequence of actions $\hat{\alpha}$ that successfully completes the specified goal. An action sequence $\hat{\alpha} = (a_1, a_2, \ldots, a_T)$ is considered successful if and only if the resulting terminal state satisfies a goal validation predicate $V$. Let $\mathcal{T}$ be the state transition function of the GUI environment, where $S_{t+1} = \mathcal{T}(S_t, a_t)$. A sequence $\hat{\alpha}$ is accepted iff

$$V(\mathcal{T}(S_0, \hat{\alpha}), g) = \texttt{pass}.$$

The agent in our framework operates using a ReAct-style (Yao et al., 2023) loop to maintain an explicit transcript of its reasoning and interactions. At step $t$, the transcript is

$$\mathcal{H}_t = \big(u_1, a_1, o_1, \, u_2, a_2, o_2, \, \ldots, \, u_t, a_t, o_t\big),$$

where $u_i$ are the model's *thoughts* (free-form reasoning), $a_i$ are *actions* (e.g., clicks or text inputs), and $o_i$ are the resulting *observations* (the new screenshot and GUI elements $(scr_i, e_i)$). The policy $\pi_\theta$ conditions on $\mathcal{H}_t$ to generate the next thought and action, $(u_{t+1}, a_{t+1}) \sim \pi_\theta(\cdot \mid \mathcal{H}_t, g)$. Executing $a_{t+1}$ yields the observation $o_{t+1} = (scr_{t+1}, e_{t+1})$, which is appended back to the transcript. This process is strictly sequential and continues until the agent executes a finish action or reaches its step budget.

### 2.2 SUPERVISION WITH A STANDARD PRM

To guide the GUI Agent towards an optimal action sequence, a supervisory signal is introduced at each step. At any given step $t$, the policy $\pi_\theta$ first generates a set of $k$ candidate thought-action pairs, $\mathcal{C}_t = \{(u_{t,j}, a_{t,j})\}_{j=1}^k$. Based on the history, the standard PRM evaluates these candidates and selects the best action to execute. It computes a reward score for each candidate thought-action $(u_{t,j}, a_{t,j})$ conditioned on the complete, unaltered interaction transcript $\mathcal{H}_{t-1}$. The model's objective is to identify the action most likely to lead to a successful trajectory. The selected action $a_t$ is formally determined by:

$$(u_t, a_t) = \operatorname*{argmax}_{(a,u) \in \{(a_{t,j}, u_{t,j})\}_{j=1}^k} \big( \text{PRM}(g, scr_{t-1}, e_{t-1}, \mathcal{H}_{t-1}, (a, u)) \big) \tag{1}$$

The corresponding thought $u_t$ is selected along with $a_t$. This approach relies on the supervisory model's ability to effectively process the raw, and potentially long, history.

## 3 GUI-PRA

In this section, we present the methodology of GUI-PRA. GUI-PRA is specifically designed to address the two core limitations of PRMs in this context: the "lost in the middle" problem with long histories, and a lack of UI changing awareness. The framework achieves this through a three-stage

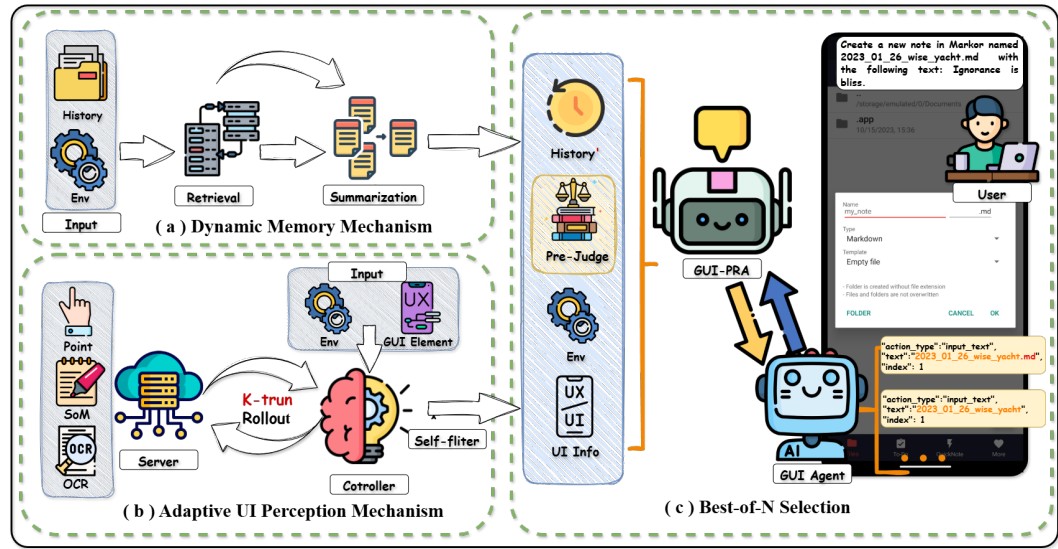

Figure 2: The overall workflow of GUI-PRA. **(a)** The Dynamic Memory module first processes the raw interaction history to generate a condensed summary. **(b)** Concurrently, the **Adaptive UI Perception Mechanism** actively reasons about the UI state to select the most appropriate tool for gathering grounded visual evidence. **(c)** For the final Best-of-N Selection, GUI-PRA integrates these two information streams along with the previous action and its score from the last step to evaluate and select the optimal candidate action.

process: (1) the **Dynamic Memory** mechanism to condense long and noisy interaction histories, (2) the **Adaptive UI Perception** mechanism to actively reason about UI state changes and gather grounded evidence, and (3) the **Best-of-N Selection** process where the outputs of the first two stages are integrated to provide an informed supervisory signal. The complete workflow is illustrated in Figure 2.

## 3.1 DYNAMIC MEMORY MECHANISM

The process reward model's performance can be degraded by long and noisy interaction transcripts, a challenge often termed the "lost in the middle" problem. To mitigate this, we introduce a Dynamic Memory Mechanism, which serves as the core of our GUI-PRA. This mechanism formalizes the memory function $f_{\mathrm{mem}}$, transforming the full transcript $\mathcal{H}_{t-1}$ into a compressed yet comprehensive summary $\mathcal{H}'_{t-1}$. The function operates via a two-stage process.

First, a Relevance-based Retrieval stage isolates the most pertinent recent interactions. We define a retrieval function, $f_{\mathrm{retrieve}}$, that takes the full transcript and identifies a relevance window of size $m$. This function filters the history to preserve only the $m$ most recent thought-action-observation tuples:

$$\mathcal{H}_{\mathrm{recent}} = f_{\mathrm{retrieve}}(\mathcal{H}_{t-1}) = \big((u_{t-m}, a_{t-m}, o_{t-m}), \ldots, (u_{t-1}, a_{t-1}, o_{t-1})\big). \quad (2)$$

The occluded, earlier portion of the transcript is denoted as:

$$\mathcal{H}_{\mathrm{early}} = \big((u_1, a_1, o_1), \ldots, (u_{t-m-1}, a_{t-m-1}, o_{t-m-1})\big). \quad (3)$$

Second, a Progressive Summarization stage condenses the high-level narrative of the early interactions while discarding low-level noise. We define a summarization function, $f_{\mathrm{sum}}$, which processes the early history and synthesizes it into a single, concise natural language sentence, $\mathcal{S}_{\mathrm{sum}}$:

$$\mathcal{S}_{\mathrm{sum}} = f_{\mathrm{sum}}(\mathcal{H}_{\mathrm{early}}). \quad (4)$$

Finally, the compressed history $\mathcal{H}'_{t-1}$ is constructed by prepending the textual summary to the sequence of recent interactions. This provides the supervisory model with a refined context that balances long-term narrative with short-term, high-fidelity details. The complete memory function is thus the composition of these two stages:

$$\mathcal{H}'_{t-1} = f_{\mathrm{mem}}(\mathcal{H}_{t-1}) = \mathrm{concat}(\mathcal{S}_{\mathrm{sum}}, \mathcal{H}_{\mathrm{recent}}). \quad (5)$$

## 3.2 Adaptive UI Perception Mechanism

A primary limitation of standard PRMs is their "state-change blindness"; they evaluate actions based on textual history, failing to perceive the visual consequences of those actions on the GUI. To overcome this, we introduce the **Adaptive UI Perception Mechanism**, which endows GUI-PRA with UI changing awareness, transforming it from a passive evaluator into an active perceiver.

At its core, this mechanism operates as a **"perceive-reason-verify" loop**. When confronted with a UI state change, GUI-PRA first perceives the change, then reasons about its nature to form a hypothesis about its informational needs. Finally, it verifies this hypothesis by intelligently selecting a tool to gather targeted visual evidence. This ensures the final evaluation is a grounded judgment, based on the most relevant real-time information.

To execute this loop, we equip the agent with two complementary server-side tools (Su et al., 2025), OmniParser and Point, which provide global and local UI perception respectively (detailed in Appendix C). The core of our adaptive mechanism lies in the hypothesis-driven selection between these tools. For instance, when a major UI change occurs (e.g., a screen transition), the agent requires a holistic understanding of the new layout and selects OmniParser. Conversely, when the task requires interacting with a specific, fine-grained detail (e.g., locating a particular icon), it selects the Point tool for precise grounding.

We formalize this hypothesis-driven process as an iterative information-gathering loop that can run for a maximum of $K$ iterations. At each sub-step $i$ (for $i = 1, \ldots, K$), the agent's tool policy, $\pi_{\text{tool}}$, makes a reasoned decision to select the next tool based on the goal $g$, the previous state $(scr_{t-1}, e_{t-1})$, the summarized history $\mathcal{H}'_{t-1}$, and all evidence gathered so far $\mathcal{I}_{i-1}$:

$$\text{tool}_i \sim \pi_{\text{tool}}(\cdot \mid g, scr_{t-1}, e_{t-1}, \mathcal{H}'_{t-1}, \mathcal{I}_{i-1}). \tag{6}$$

The policy can also output a special `Terminate` tool if it deems that sufficient information has been gathered. The loop continues until the `Terminate` tool is used or the limit $K$ is reached. The sequence of tool outputs, $\mathcal{I}_n = \{\text{UI}_1, \ldots, \text{UI}_n\}$, is then synthesized by an aggregation function, $f_{\text{aggregate}}$, into the final, refined evidence $\text{UI}_t = f_{\text{aggregate}}(\mathcal{I}_n)$. The complete tool-use function is thus defined as:

$$g_{\text{tool}}(g, scr_{t-1}, e_{t-1}, \mathcal{H}'_{t-1}) = \text{UI}_t.$$

## 3.3 Selection: Fine-grained Reward Scoring

After processing the task context through the Dynamic Memory and Adaptive UI Perception mechanisms, the GUI-PRA framework successfully assembles a condensed yet relevant action history, $\mathcal{H}'_{t-1}$, grounded visual evidence, $\text{UI}_t$. The subsequent and critical task is to leverage this synthesized information to accurately and efficiently score a set of candidate actions, thereby enabling a Best-of-N selection strategy. To this end, we have designed a comprehensive **Scoring Mechanism** with the following key features. The complete prompt designed for this scoring mechanism is detailed in Appendix F.3.

**Fine-grained Scoring Scale.** We establish a scoring range from 0 to 10, which is partitioned into five distinct tiers (e.g., 0-2, 3-4, etc.). To guide the model toward nuanced and detailed judgment, we provide a thorough description and explicit criteria for each scoring tier. This structured rubric ensures that the model can perform a fine-grained evaluation of each candidate action.

**Explicit Penalty Rules.** To suppress ineffective exploration, the scoring mechanism incorporates clear penalty clauses. When the GUI Agent executes a repetitive or demonstrably incorrect action, the GUI-PRA deducts points according to these rules. This negative feedback effectively steers the agent away from such behaviors in future steps.

**Contextual Consistency.** To maintain objectivity and ensure consistency across consecutive steps, we introduce a contextual reference mechanism. Specifically, when scoring the candidate actions for the current turn, we include the action selected in the previous turn along with its final score as part of the input. This allows the GUI-PRA to base its scoring not only on the current state but also on its own recent evaluations, ensuring the reward signal is both stable and temporally consistent.

### 3.4 Enhanced Supervision with GUI-PRA

In contrast to the standard supervision process metioned in § 3.4, which relies on raw, unprocessed inputs, our GUI-PRA framework provides **Enhanced Supervision** through two key architectural modifications. These enhancements transform the supervisory signal from being static and context-agnostic to dynamic and well-grounded.

First, whereas a standard PRM conditions its evaluation on the complete and often noisy interaction transcript $\mathcal{H}_{t-1}$, GUI-PRA utilizes a refined historical context. It employs the dynamic memory function, $f_{\text{mem}}$, to generate a concise and salient summary, $\mathcal{H}'_{t-1} = f_{\text{mem}}(\mathcal{H}_{t-1})$. This allows the supervisory model to focus on the most relevant prior steps, mitigating the "lost in the middle" problem.

Second, and more critically, GUI-PRA directly confronts the standard PRM's lack of UI changing awareness. A standard PRM provides static evaluations disconnected from the visual consequences of actions. To overcome this critical mismatch, our Adaptive UI Perception mechanism ($g_{\text{tool}}$) provides dynamic, real-time visual evidence, $\text{UI}_t$. This evidence, gathered by actively reasoning about UI changes, serves to ground the evaluation in the current visual reality of the task.

Consequently, the final action selection is conditioned on both a focused history and grounded visual feedback, making the decision significantly more informed. This enhanced, multimodal supervision process is formalized as follows:

$$(u_t, a_t) = \operatorname*{argmax}_{(a,u) \in \{a_{t,j}, u_{t,j}\}_{j=1}^{k}} \left( \text{GUI-PRA}(g, scr_{t-1}, e_{t-1}, \mathcal{H}'_{t-1}, (a, u), \text{UI}_t) \right) \tag{7}$$

## 4 Experiment

### 4.1 Benchmark

We choose two online Mobile benchmark and invloves M3A (Rawles et al., 2025) as our excution environment, a zero-shot framework that integrates ReAct and Reflexion principles, processing Set-of-Mark (SoM) annotated screenshots to generate structured JSON actions.

**AndroidWorld** (Rawles et al., 2025) is a dynamic benchmark for GUI agents developed for the Android ecosystem. It spans 116 tasks across 20 real-world applications. The benchmark establishes a realistic, online environment by leveraging Android Studio, specifically emulating a Pixel 6 device model running Android 13 (API Level 33). A key feature of AndroidWorld is its use of task templates, where specific task instances are generated and controlled via random seeds, ensuring reproducibility. The tasks are categorized into three difficulty levels—easy, medium, and hard—allowing for a more granular evaluation of an agent's capabilities.

**MobileMiniWoB++** is a mobile-centric web benchmark adapted by Rawles et al. (2025) from the original MiniWoB++ benchmark (Liu et al., 2018). It comprises 92 tasks, all of which are integrated within a single simulated application, meaning the tasks do not involve multi-page navigation. Consistent with traditional web benchmarks, the tasks in Mobile-MiniWoB++ typically feature a high density of UI elements, presenting a significant challenge to the agent's element localization abilities. A notable limitation of this benchmark is that its task templates are not fully controllable, leading to minor variations in the specific details of each task instance.

### 4.2 Baselines

Our GUI-PRA framework is designed to transform a standard Process Reward Model (PRM) into a domain-specific supervisor. Consequently, we evaluate its performance against two primary baselines:

- **Base Agent (No Guidance):** A standalone GUI agent operating without any external supervision. This baseline measures the raw capability of the base model.

- **Standard PRM Guidance:** The same base agent guided by a standard, powerful PRM. This baseline isolates the benefit of our GUI-specific enhancements over a generic guidance method.

Table 1: Main performance comparison on AndroidWorld and MobileMiniWoB++. The guidance backbones are denoted by **-I** (InternVL3-78B-Instruct) and **-Q** (Qwen2.5-VL-72B-Instruct). $\Delta$@**1**: SR% gain over the base model; $\Delta$@**2**: SR% gain of GUI-PRA over the standard PRM.

| Model Series | Setting | DSR (%) | | | SR (%) | $\Delta$@1 | $\Delta$@2 |
|---|---|---|---|---|---|---|---|
| | | easy | medium | hard | | | |
| *AndroidWorld* | | | | | | | |
| InternVL3 | INTERNVL3-8B-INSTRUCT | 9.84 | 0.00 | 5.26 | 6.03 | - | - |
| | W/ PRM-I | 16.39 | 0.00 | 5.26 | 9.48 | +3.45 | - |
| | W/ GUI-PRA-I | **26.23** | 1.39 | 5.26 | **15.09** | +9.06 | +5.61 |
| Qwen2.5-VL | QWEN2.5-VL-7B-INSTRUCT | 18.85 | 2.78 | 5.26 | 11.64 | - | - |
| | W/ PRM-Q | 32.79 | 2.78 | 5.26 | 18.97 | +7.33 | - |
| | W/ GUI-PRA-Q | 32.79 | **9.72** | 5.26 | **21.12** | +9.48 | +2.15 |
| Mixture Models | INTERNVL3_5-8B-INSTRUCT | 11.48 | 0.00 | 5.26 | 6.90 | - | - |
| | W/ PRM-Q | 31.15 | 2.78 | 5.26 | 18.10 | +11.20 | - |
| | W/ GUI-PRA-Q | 31.15 | 2.78 | 5.26 | 18.10 | +11.20 | 0.00 |
| *MobileMiniWob++* | | | | | | | |
| InternVL3 | INTERNVL3-8B-INSTRUCT | - | - | - | 39.13 | - | - |
| | W/ PRM-I | - | - | - | 36.96 | -2.17 | - |
| | W/ GUI-PRA-I | - | - | - | **42.39** | +3.26 | +5.43 |
| Qwen2.5-VL | QWEN2.5-VL-7B-INSTRUCT | - | - | - | 38.04 | - | - |
| | W/ PRM-Q | - | - | - | 47.82 | +9.78 | - |
| | W/ GUI-PRA-Q | - | - | - | **57.61** | +19.57 | +9.79 |

## 4.3 EXPERIMENTAL SETUP

**Model Selection.** We conduct experiments using models from two prominent series: Qwen2.5-VL (Bai et al., 2025) and InternVL3 (Zhu et al., 2025a).

**Agent and Supervisor Roles.** For the role of the base GUI Agent, we utilize the moderately-sized Qwen2.5-VL-7B-Instruct and InternVL3-8B-Instruct. For the supervisory role in both the Standard PRM baseline and our GUI-PRA framework, we employ their larger, more powerful counterparts: Qwen2.5-VL-72B-Instruct and InternVL3-78B-Instruct.

**Cross-Family Generalization Setting.** To assess the generalization capabilities of the supervisory models, we also evaluate a mixed-model setting where the InternVL3-8B-Instruct agent is guided by the Qwen2.5-VL-72B-Instruct supervisor.

## 4.4 EVALUATION METRICS

To provide a comprehensive assessment of our method, we employ two key metrics that evaluate both the effectiveness and the efficiency of our GUI-PRA.

**Success Rate (SR).** This is the primary metric for measuring the overall effectiveness of the agent. It is defined as the percentage of tasks that the agent successfully completes out of the total number of trials. A higher SR directly corresponds to a more capable and reliable agent.

**Difficulty-Stratified Success Rate (DSR).** To provide a more granular analysis of agent capabilities, we introduce the Difficulty-Stratified Success Rate (DSR). This metric disaggregates the overall Success Rate (SR) to report separate performance scores for tasks classified as 'easy', 'medium', and 'hard'. This breakdown pinpoints the specific task complexities where our framework delivers the most value.

## 4.5 RESULTS AND ANALYSIS

**Overall Performance Superiority.** As shown in Table 1, our GUI-PRA framework consistently delivers superior performance over both the unguided base models and those guided by a standard PRM. On the AndroidWorld benchmark, GUI-PRA boosts the Qwen2.5-VL model's overall success

Table 2: **Ablation study of GUI-PRA components.** Performance is reported for the series full model based on Qwen2.5-VL and variants with key components removed. The $\triangle@2$ column shows the SR% gain over the standard PRM. The full model performs best, showing the value of each component.

| Method | DSR (%) | | | SR (%) | $\triangle@2$ |
|---|---|---|---|---|---|
| | easy | medium | hard | | |
| **GUI-PRA (Full)** | **32.79** | **9.72** | **5.26** | **21.12** | **+2.15** |
| *w/o component:* | | | | | |
| — OmniParserV2 | 32.79 | 2.78 | 5.26 | 18.97 | 0.00 |
| — Point | 29.51 | 6.94 | 5.26 | 18.53 | -0.44 |
| — Memory | 31.15 | 0.00 | 5.26 | 17.24 | -1.73 |

rate (SR) by 9.48%. This performance advantage is even more pronounced on the general-purpose InternVL3 model, where GUI-PRA provides a much larger improvement margin, elevating the SR from 6.03% to 15.09% (+9.06%). This trend extends to the UI-dense Mobile-MiniWoB++ benchmark, where GUI-PRA achieves an impressive 19.57% SR gain for Qwen2.5-VL, more than doubling the performance boost offered by the standard PRM (+9.78%).

**Critical Advantage in Complex Tasks.** A more granular analysis using the Difficulty-Stratified Success Rate (DSR) reveals that GUI-PRA's most significant advantages emerge on tasks of 'medium' difficulty. For the InternVL3 series, which completely fails on these tasks (0.00% SR) both standalone and with a standard PRM, GUI-PRA is the only method that enables a non-zero success rate (1.39%). The impact is even more substantial on the more capable Qwen2.5-VL model, where GUI-PRA elevates the 'medium' task success rate from 2.78% to 9.72%, more than tripling the performance. This demonstrates that GUI-PRA provides a critical boost on moderately challenging problems, unlocking capabilities for weaker models and substantially enhancing them for stronger ones.

### 4.6 ABLATION STUDIES

To validate the distinct contributions of our core mechanisms, we conducted an ablation study on the Qwen2.5-VL series. As presented in Table 2, the results confirm that each component is critical. The full model achieves the best performance, while removing any single mechanism leads to a significant degradation.

**Dynamic Memory is the pillar for contextual understanding.** Our memory module is not a passive store but an active filtering and summarization mechanism. Removing it forces the agent to contend with raw, unfiltered history, causing a catastrophic performance collapse on context-dependent tasks; the success rate on 'medium' difficulty tasks plummets from 9.72% to zero. This confirms that *processed, high-signal memory*—not just the presence of history—is indispensable for solving complex tasks.

**Adaptive UI Perception is critical for grounded judgment.** This mechanism's value is evident when its perceptual tools are removed. Removing the global context from OmniParser nullifies any advantage over the standard PRM on 'medium' tasks. More revealingly, removing the local grounding from the Point tool causes performance to drop *below* the standard PRM baseline (-0.44%). This outcome demonstrates a critical insight: our framework's advanced reasoning becomes a liability without its perceptual tools. GUI-PRA is designed to form hypotheses and expect verification; when the verification step fails, its sophisticated judgment becomes miscalibrated. This leads to flawed evaluations that are more detrimental than the simpler, static judgments of a standard PRM. The full model's success, therefore, relies on the tight integration of processed memory and an active, multi-level perception system.

### 4.7 CASE STUDIES

Figure 3 illustrates a complete operational flow of our GUI-PRA framework, showcasing its interaction with a base GUI Agent to fulfill a user's request. The top row depicts the trajectory of the base GUI Agent. It correctly executes all the positive data entry sub-tasks: it navigates to the contacts

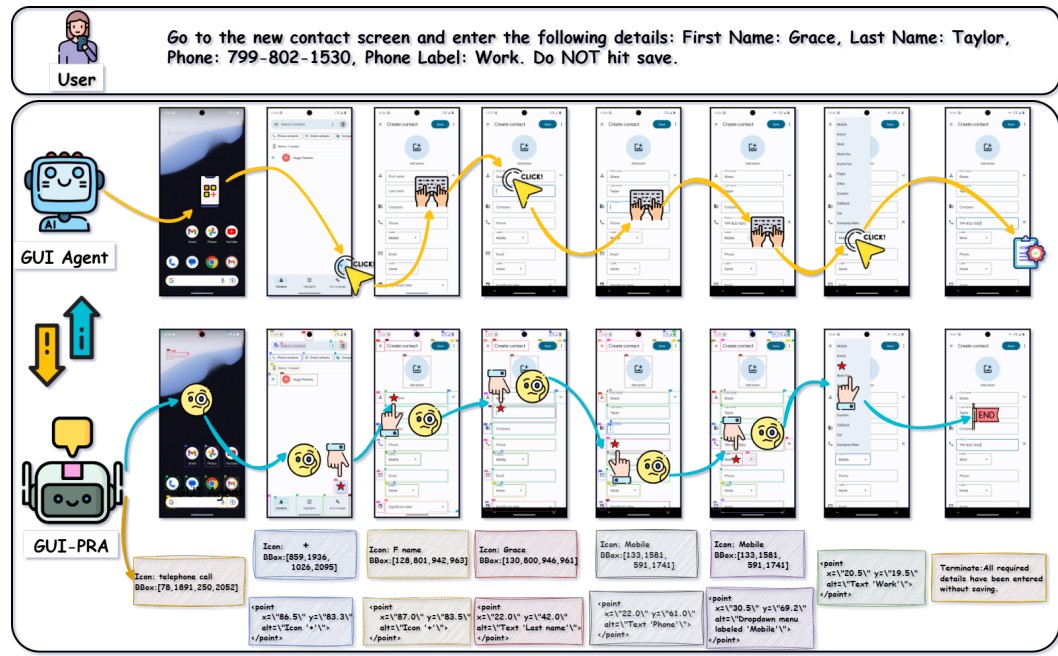

Figure 3: A complete case of GUI-PRA guiding a GUI Agent to complete the 'ContactsNewContactDraft' task. The figure illustrates the parallel process flows, showing the agent's action trajectory (top row) and the continuous supervision provided by GUI-PRA (bottom row) across multiple steps until task completion.

application, initiates the creation of a new contact, and accurately inputs the name, phone number, and label. In parallel, the bottom row shows the continuous monitoring and reasoning process of our GUI-PRA. At each step, GUI-PRA leverages its UI Tools to perceive the screen, grounding the agent's actions and the state of the UI elements, as evidenced by the 'Icon' and '<point>' outputs. The primary challenge here is not the data entry itself, but correctly interpreting the negative constraint: "Do NOT hit save." GUI-PRA excels by continuously validating the agent's progress against this complex goal. It correctly determines the precise moment the task is finished, instructing the agent to terminate rather than incorrectly proceeding to save.

This intervention prevents the GUI Agent from making an irreversible error that would have resulted in task failure. This case highlights GUI-PRA's ability to provide nuanced, process-level supervision that goes beyond simple action validation, ensuring strict adherence to complex user constraints.

Furthermore, we observe that GUI-PRA's scoring feedback mechanism and its penalty for repeated actions were instrumental in guiding it to this correct decision, as detailed in Appendix D.

## 5 CONCLUSION

In this paper, we introduced GUI-PRA, a novel, training-free framework that transforms a standard Process Reward Model (PRM) into a GUI-domain-specific supervisor. Our work addresses two critical limitations of standard PRMs in dynamic GUI environments: the "lost in the middle" phenomenon with long-context histories, and the lack of UI changing awareness that leads to static evaluations. To overcome these challenges, GUI-PRA incorporates two core innovations. A Dynamic Memory mechanism intelligently condenses historical trajectories to maintain focus on salient information. More critically, an Adaptive UI Perception mechanism endows the agent with UI changing awareness, enabling it to reason about visual changes and gather grounded evidence before making a judgment. Extensive experiments on online GUI benchmarks validate the efficacy of our approach, showing that GUI-PRA significantly improves agent success rates, particularly on more challenging d tasks. This highlights its potential to robustly enhance the reliability and efficiency of automated GUI agents in dynamic environments.

## ETHICS STATEMENT

Our GUI-PRA framework significantly enhances the autonomy and reliability of GUI agents, making them more capable of executing complex tasks in real-world digital environments. While the ability to automate complex digital interactions is a powerful tool, it also introduces potential risks. A highly autonomous agent could be misused for malicious purposes, such as unauthorized data access, spam generation, or performing actions without explicit user consent. We strongly urge researchers and developers to implement robust safety protocols, such as clear user consent mechanisms and operational constraints, to ensure the ethical deployment of such technologies. Nevertheless, the original goal of our work is positive: to create more helpful and efficient digital assistants that can robustly follow user instructions. Therefore, we encourage the community to leverage this technology responsibly, with a focus on beneficial and user-centric applications.

## REPRODUCIBILITY

To ensure the reproducibility of our findings, detailed implementation parameters and prompts can be found in Appendix B. Additionally, our key source code has been submitted as part of the supplementary material. These measures are intended to facilitate the verification and replication of our results by other researchers in the field.

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

## A STATEMENT ON THE USAGE OF LARGE LANGUAGE MODELS

During the preparation of this manuscript, a Large Language Model (LLM) was utilized as an auxiliary tool. Its application was strictly limited to improving the language and readability of the text, as well as assisting with the formatting of figures. The authors have meticulously reviewed and edited all machine-generated suggestions to ensure the scientific accuracy and integrity of the final content, for which they take full responsibility.

## B IMPLEMENTATION DETAILS

To ensure the reproducibility of our experimental results, we meticulously documented and controlled several key parameters and settings throughout the evaluation of GUI-PRA.

For the underlying GUI Agent, we fixed the base inference parameters to maintain consistent behavior across all experiments. Specifically, the temperature was set to $0.5$, top_p to $0.9$, and top_k to $80$. We use random seeds to control generation process. During the base model testing phase, a random seed of $42$ was used. For test-time scaling experiments, where eight candidate trajectories were generated, the following distinct random seeds were employed: $[30, 42, 3407, 114514, 256, 64, 1024, 2]$.

In the configuration of GUI-PRA's components, the activation threshold for the dynamic memory mechanism was set to $5$, triggering its use when the historical record length exceeded five steps. The maximum number of routing attempts for the dynamic UI Tool Routing component was capped at $2$.

The experiments were conducted on the following hardware configurations: 4x H20 GPUs with 96GB VRAM, 1x A100 GPU with 40GB VRAM, 2x L20 GPUs with 48GB VRAM.

## C TOOL DETAILS

Table 3: The perceptual UI Tools used by GUI-PRA for interface analysis.

| Tool | Input | Output | Description |
|------|-------|--------|-------------|
| OMNIPARSER | image | SoM + BBox | text-driven object detection |
| POINT | image + description | point coordinates | object localization |

The Adaptive UI Perception mechanism of GUI-PRA is facilitated by two complementary, server-side tools. Their input/output formats are summarized in Table 3, and their specific functionalities are detailed below:

**OmniParser: Global UI Perception**. The OmniParser tool (Lu et al., 2024) is designed for comprehensive GUI interface recognition. Its process consists of two primary stages: Optical Character Recognition (OCR) and Set-of-Mark (SoM) annotation. First, the OCR module interprets the semantics of various elements on the GUI and precisely localizes their bounding boxes. Following this, the Set-of-Mark module utilizes the content and coordinates from the OCR stage to precisely annotate the interface. This yields both a structured textual representation and an intuitive visual overlay of the interface, both of which are readily interpretable by the Large Language Model (LLM).

**Point: Local UI Element Grounding**. The Point tool, based on Molmo-7B-D-0924 (Deitke et al., 2024), is engineered to precisely ground UI elements from natural language descriptions. It can locate the coordinates of a UI element based on its corresponding textual content or identify the position of common GUI icons from more ambiguous, descriptive prompts (e.g., "Phone Icon"). To provide a clear visual representation for the PRM, we overlay the original GUI screenshot with a red pentagram at the coordinates generated by the Point tool, effectively highlighting the targeted element.

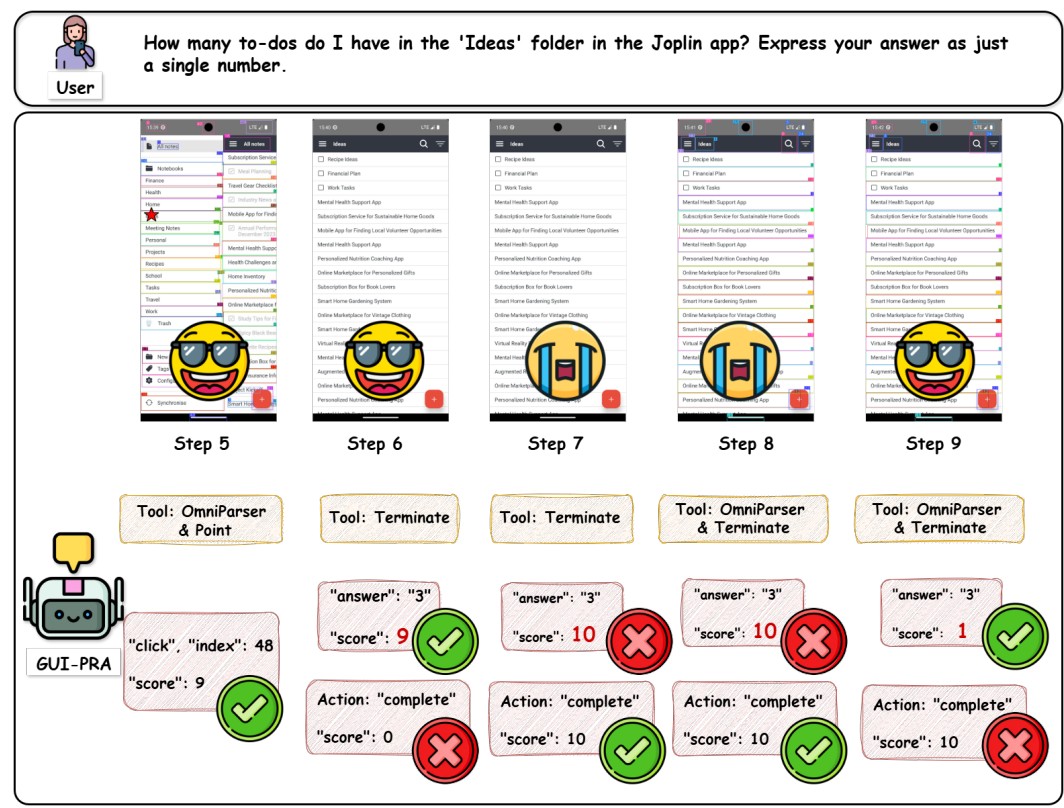

Figure 4: A case study illustrating GUI-PRA's self-correction from an evaluation loop. The figure shows GUI-PRA assigning conflicting high scores to both the correct answer and a premature termination action (Steps 7-8), before correcting its judgment in Step 9 to successfully guide the agent to task completion.

## D  CASE STUDY: PENALTY FOR REPEATED ACTIONS

The case study in Figure 4 demonstrates a critical capability of GUI-PRA: its ability to self-correct after entering a flawed evaluation loop. The user's objective is for the agent to count the to-do items on the screen and provide a numerical answer.

The sequence shows the agent successfully navigating to the correct "Ideas" screen (Step 6), where the answer is visually available. However, a problem arises in the evaluation process. In Steps 7 and 8, GUI-PRA incorrectly gives a perfect score of 10 to both the correct intermediate action ('"answer": "3"') and the premature final action ('Action: "complete"'). This creates a conflicting signal, trapping the process in a non-productive cycle because it endorses two contradictory steps as equally valid.

The crucial intervention occurs in Step 9. Here, GUI-PRA breaks the stalemate by correcting its own flawed judgment. It now correctly penalizes the repetitive and premature 'complete' action while validating the 'answer' action as the correct path forward. This decisive re-evaluation resolves the ambiguity, breaks the loop, and guides the agent to successfully complete the task by providing the final answer.

## E  RELATED WORK

Recently, GUI agents powered by (Multimodal) Large Language Models ((M)LLMs) have demonstrated significant potential in Graphical User Interfaces (GUIs) automating tasks. Despite these advancements, existing GUI agents still face challenges in completing complex online GUI tasks. To address these limitations, many researchers have attempted to decompose the core capabilities of a GUI agent, such as planning and grounding, to design more sophisticated agent frameworks (Ye et al.,

2025; Zhang et al., 2023). For example, **Mobile-Agent-v3** (Ye et al., 2025) involves the coordination of multiple GUI agent roles that share observations and reasoning trajectories to handle complex, long-horizon automation workflows. Another line of research focuses on building GUI-specific agents through fine-tuning (Liu et al., 2025; Hong et al., 2024; Wu et al., 2024; Li et al., 2025). For instance, InfiGUI-R1 (Liu et al., 2025) employs a two-stage reinforcement learning paradigm to enhance an agent's spatial reasoning and error recovery capabilities, respectively. However, a commonality in these existing methods is their reliance on the agent itself making the correct decision at each individual step. This dependency increases the risk of task failure, especially when an irreversible action is taken. In response to this challenge, we introduce a Process Reward Agent for GUI tasks, which leverages an external agent to provide process supervision, thereby pre-evaluating and selecting more optimal execution paths.

### E.1 PROCESS REWARD MODELS FOR LLMs

Techniques such as Chain-of-Thought (CoT) (Wei et al., 2023) and Chain-of-Action (CoA) are designed to help LLMs deconstruct complex problems into a sequence of manageable steps for thought or action. However, during long-chain reasoning processes, LLMs do not always generate logically sound steps and may even produce self-contradictory outputs. Some existing works have explored self-reflection (Shinn et al., 2023; DeepSeek-AI et al., 2025) and sef-refine (Madaan et al., 2023; Pan et al., 2024; Tyen et al., 2024) mechanisms to rectify these reasoning errors. Yet, the efficacy of such methods is often constrained by the intrinsic capabilities of the model itself, leading to low success rates or causing the model to become trapped in inefficient correction loops. In contrast, an alternative and often more effective approach is to introduce external supervision. Several studies (Gandhi et al., 2025; Xiong et al., 2025; Wanyan et al., 2025; Xiao et al., 2025) have proposed the use of a Process Reward Model (PRM) to provide external oversight and feedback on the LLM's reasoning process, helping it select the optimal reasoning path. In the GUI agent domain, works like Hu et al. (2025b); Wanyan et al. (2025) have constructed PRMs using reinforcement learning techniques. However, these methods typically demand rigorous data preparation and entail significant training overhead. Distinguishing our work from these training-intensive approaches, we transform a standard PRM into a GUI-specific Process Reward Agent (PRA) by designing a novel training-free Judge Agent framework.

## F PROMPTS

We provide the prompts in constructing GUI-PRA below.

### F.1 GUI-PRA: MEMORY

---

**GUI-PRA: Dynamic Memory - Stage 1**

**SYSTEM:**

You are a **Process Reward Model**. Your task is to evaluate a single **candidate action step** based on a **user's prompt** and **provided screen image**. To reduce the impact on the dialogue window, you need to dynamically manage the cache. Please dynamically manage the **user's action history** part, keeping only the necessary portions. Ensure that the essential key information is retained.

**CRITICAL RULES:**

1. You MUST return a list of the EXACT SAME LENGTH as the input history

2. You MUST only keep the last N recent steps (where N is determined by relevance)

3. You MUST set all non-essential earlier steps to empty strings "

4. You MUST NOT skip steps or create gaps - only preserve consecutive recent steps from the end

5. You MUST maintain the original step numbering and format

**Selection Criteria:**

---

- Preserve only the most recent steps necessary for current context
- Remove redundant or outdated information from the beginning
- Keep steps that provide essential operational context
- Consider both textual content and visual context from the screen image

**Output Format:** Return **ONLY** a Python list with the same length as input, where unwanted steps are empty strings.

**USER:**

Current Goal: {goal}
Full History (as list): {history}
Task: Return a filtered list of the **SAME LENGTH** where only the **last N relevant steps** are preserved (as-is) and all earlier steps are set to empty strings.
Example Input: ['Step 1 -A', 'Step 2 -B', 'Step 3 -C', 'Step 4 -D']
Example Output: ['', '', 'Step 3 -C', 'Step 4 -D']
Return ONLY the Python list format, nothing else.

---

**GUI-PRA: Dynamic Memory - Stage 2**

**SYSTEM:**
You are a helpful assistant that summarizes text.
**USER:**
You are an expert summarizer. Your task is to read a list of previous user actions and create a concise, **one sentence** summary. The summary should capture the main accomplishments and the state reached before the final few steps. Actions to Summarize: {actions} Instructions:

- Be concise and to the point.
- Write in a narrative style (e.g., "The user logged in and navigated to...").
- Do not use a list format or mention step numbers.
- The summary should provide context for the "Recent Actions" that will follow it.

Output: Provide ONLY the summary sentence.

## F.2 GUI-PRA: UI TOOL ROUTING

**GUI-PRA: UI Tool Routing**

**SYSTEM:**
You are a visual assistant with the ability to collect external information using different tools, specifically for tasks involving Computer, Phone, and Browser Use judging. Your goal is to evaluate the type of problem based on the input question and choose the most appropriate tool to gather relevant information for a subsequent process reward model to judge the response. You only need to decide to use the listed tools to enhance your understanding of the question, not to answer it.

Here are the available tools:

- **Point**: Identifies a specific point... Example:
  ```
  {"name": "Point", "arguments": {"image": "img_1", "param": "Icon 'Gmail'"}}
  ```
- **omni_parser**: Parses a UI or general image... Example:
  ```
  {"name": "omni_parser", "arguments": {"image": "img_1"}}
  ```

- **Terminate**: Ends the task and provides... Example:

  ```
  {"name": "Terminate", "arguments": {"ans": "1985"}}
  ```

To gather relevant information:

- Assess the type of question provided...
- If segmentation or line drawing is required, first use the **Point** tool to identify coordinates.
- Use the selected tools logically and sequentially...

Always ensure that at least one tool is used, and structure the output in a JSON format as shown below:

**Example Output:**
*Example 1:*

```
{
"thought":  "My primary objective is to gather sufficient
information to score the next action for a Process
Reward Model (PRM). To do this, I need a comprehensive
understanding of the entire screen, including all text and
interactive elements.  The omni_parser tool is the most
effective choice as it provides a complete analysis of
the UI. Therefore, I will use it to collect the necessary
context for the evaluation.",
"actions":  [
{"name":  "omni_parser", "arguments":  {"image":  "img_1"}}
]
}
```

*Example 2:*

```
{
"thought":  "In order to help to evaluate the next action
for the PRM, I need to gather the necessary information
first.  The next action is likely related to the weather
information, identified by the text 'Sun, Oct 15'.
Therefore, I must pinpoint its location.  I will use the
Point tool to obtain the coordinates of this text.",
"actions":  [
{"name":  "Point", "arguments":  {"image":  "img_1",
"param":  "Text 'Sun,Oct 15'"}}
]
}
```

If further action is required, continue building on the previous step with the correct tool.

---

## GUI-PRA: UI Tool Routing

**USER:**
User Question: *<initial_prompt>*
You have already taken some steps. Here is the history of your actions and their observations:
Current tool calling history: *<history_str>*

**Your Task (OI - Observation & Introspection):**
**Summarize**: Briefly summarize what you have learned from the history.

**Decide**: Based on your summary and the initial goal, decide on the next step. Do you have enough information to answer the request?

- **If YES**, call the tool: `{"name": "Terminate", "arguments": {"ans": "<your final answer>"}}`

- **If NO**, call another tool to get the missing information.

**Do not call any tool that you have used before.**

### F.3 GUI-PRA: BEST OF N SELECTION

#### GUI-PRA: BoN Selection

**SYSTEM:**

You are a Process Reward Model (PRM). Your task is to evaluate a single candidate action step based on a user's instruction, a provided screen image, and other contextual information. Do not give a high score just because the reason and the action within the response are consistent. You need to prioritize whether the action is performed correctly.

**Evaluation Process and Criteria:**

1. Understand the Goal and Context: Carefully review the user's final objective, the current screen image, and the history of prior actions, including previous steps.

2. Determine Your Optimal Action: Based on all available information, internally decide what the most effective and optimal next action should be to accomplish the task.

3. Evaluate the Candidate Action: Compare the provided candidate action against your optimal action, using the following detailed criteria for a comprehensive assessment:

4. Progress Toward Goal: Does the action clearly and tangibly advance the task? Reward meaningful progress; penalize irrelevant or low-impact actions.

5. Error and Stability: Did the action cause an error? Penalize based on severity (fatal errors should receive the lowest scores, while minor/recoverable errors receive smaller penalties). The score should also be reduced if the model's output is ambiguous or unstable.

6. Efficiency: Is this an efficient path to the goal? Penalize redundant or repetitive actions that yield no significant progress.

7. Reflection Usage: Does the action demonstrate learning from past mistakes (utilizing reflection)? Reward the effective use of reflection; penalize ignoring its insights.

8. Loop Detection: Does this action create a repetition or loop when compared to previous steps? Identify and penalize ineffective loops.If there are consecutive repetitive steps, please reduce the score significantly.

9. Contextual Awareness: Is the action aligned with the overall PlanningStep and TaskStep? Ensure consistency with the strategy and penalize deviations.

10. Comprehensively evaluate the correctness of the response based on the entire action history. Ensure the task is actually completed before choosing to end.

**Assign a Score:** Based on the evaluation above, assign a numerical score from 0 to 10 to the candidate action. Scoring Guidelines (0-10 Scale):

- - 9-10: Clearly advances the goal; highly efficient; strong use of reflection; no loops.

- - 7-8: Good progress; minor inefficiencies; clear use of reflection; minimal loop risk.

- - 5-6: Moderate progress; limited efficiency; moderate use of reflection; mild repetition risks.

- - 3-4: Poor progress; inefficient; weak use of reflection; noticeable loop risks.

- - 1-2: Minimal progress; repetitive actions leading to loops; significant errors or deviations from the plan.

- - 0: Severe issues: explicit loops, critical errors that block progress, wrong Action Space, or complete irrelevance to the task.

**Output Format:** Your output must be a single JSON object containing a `"score"` (as a number from 0 to 10) and the `"original_step"` (the exact text of the candidate action you evaluated). Enclose your entire JSON output within `\n<eval><\/eval>\n` XML tags.

---

**GUI-PRA: BoN Selection**

**USER:**

Please evaluate the following candidate action based on the user's instruction and the provided screen image, following all guidelines from the system prompt.
**User's Instruction:** {action_prompt}
**Candidate Action to Evaluate:** {action}
Please complete a granular scoring for the current step based on the previous steps and scores. Here's the last action and its score :previous
Your evaluation should be a JSON object with `"score"` and `"original_step"`, wrapped in `\n<eval><\/eval>\n` tags.

---

## F.4 PRM: BEST OF N SELECTION

**PRM: BoN Selection**

**SYSTEM:**

You are a **Process Reward Model**. Your task is to evaluate a single **candidate action step** based on a **user's instruction** and **provided screen image**.
**Evaluation Process:**

1. Understand the Goal: Carefully review the user's instruction and the current screen image.

2. Determine Your Optimal Action: Based on the instruction and image, decide what you believe is the best possible action step.

3. Evaluate the Candidate Action: Compare the provided candidate action step against your optimal action.

4. Assign a Score: Assign a numerical score to the candidate action from 0 to 100. If the candidate action is correct and has a correct reasoning process, a higher score should be given.

**Output Format:** Your output must be a single JSON object containing a `"score"` (as a number from 0 to 10) and the `"original_step"` (the exact text of the candidate action you evaluated). Enclose your entire JSON output within `\n<eval><\/eval>\n` XML tags.

---

**PRM: BoN Selection**

**User:**

Please evaluate the following candidate action based on the user's instruction and the provided screen image, following all guidelines from the system prompt.
**User's Instruction:** {action_prompt}
**Candidate Action to Evaluate:** {action}
Your evaluation should be a JSON object with `"score"` and `"original_step"`, wrapped in `\n<eval><\/eval>\n` tags.

