# OpenReview forum: "GUI-PRA: Process Reward Agent for GUI Tasks"
_ICLR.cc/2026/Conference — ICLR 2026 Conference Withdrawn Submission_

### Official Review · Reviewer_N483 · 2025-10-25

**Soundness:** 2
**Presentation:** 1
**Contribution:** 1
**Rating:** 2
**Confidence:** 3

**Summary:**

The authors want to improve the GUI agent in a training-free process, thus proposed GUI-PRA, a multi-step process that first uses a judge agent that guides the reward better, and deploys a visual verification module for "perceive-reson-verify" on UI actions, and finally create a selection mechanism to enhance the results. The study evaluated primarily on 2 VLM agent models and 2 benchmarks.

**Strengths:**

The tasks of interest is valuable and the combinations of techniques and sensible results make this a worthy study.

**Weaknesses:**

1. Citation format is not consistent
2. Plots are too small almost impossible to read in print. I urge the authors to revise to a legible quality.
3. Using a 3rd-party judge, or "dynamic memory mechanism" that is a relevance-based module, is widely used already. I don't recall the exact reference, but I was aware of those a long time ago.
4. The so-called "lost in the middle" or maybe "needle in the haystack" is a long-standing problem. I wonder if the author's methods can tackle LLM too, not just "agents."
5. I am thinking of an extended literature review that bridges what was done in LLM/VLM in a similar context, and then agents may be needed.
6. The experiments lack some ablations on model sizes, and if resources allowed, some closed-source models as well, since it does not require training.
7. Consider citing below as references to the adaptive visual component (sec3.2) since the motivation looks similar
[1] Hu, Yushi, et al. "Visual program distillation: Distilling tools and programmatic reasoning into vision-language models." CVPR 2024
[2] Chiu, et al. "AIDE: Agentically Improve Visual Language Model with Domain Experts" ICLR 202

**Questions:**

1. the so called "lost in the middle" or maybe "needle in the haystack" is a long standing problem, I wonder if the authors method can tackle LLM too, not just "agents."?
2. Usually those models has multiple sizes, how does the authors' methods work for other sizes?

---

### Official Review · Reviewer_p6sm · 2025-10-31

**Soundness:** 3
**Presentation:** 3
**Contribution:** 3
**Rating:** 4
**Confidence:** 2

**Summary:**

This paper proposes GUI‑PRA (Process Reward Agent for GUI Tasks), a training‑free framework designed to enhance GUI agents through improved process supervision. It introduces two key components—Dynamic Memory, which mitigates the “lost in the middle” issue by condensing long interaction histories, and Adaptive UI Perception, which enables awareness of visual state changes. Experiments on AndroidWorld and Mobile‑MiniWoB++ benchmarks show an average success rate improvement of 14.53%, outperforming the standard PRM baseline (8.56%).

**Strengths:**

- The paper presents a comprehensive GUI agent framework that integrates multiple components, demonstrating a high level of innovation and thoughtful system design.
- It evaluates the proposed approach using two different model series, which effectively supports the generalization and robustness of the method.
- The overall presentation is clear and well‑structured, with logically coherent arguments and fluent writing that make the work easy to follow.

**Weaknesses:**

- The paper lacks quantitative validation or explanation of its core motivation. While it repeatedly mentions the “lost in the middle” problem, there is no clear evidence showing how severe this issue is, how much it affects performance, or to what extent the proposed method actually mitigates it. A more detailed analysis or empirical verification would strengthen the motivation.
- The experimental evaluation is limited — only two benchmarks are used. This limited scope makes it difficult to fully assess the proposed method’s superiority. Including more baselines or benchmarks would provide a stronger evidence.
- Some figures such as Figure 3 contain text that is too small to read clearly, which negatively affects the readability and overall understanding of the paper.

**Questions:**

- In Table 1, there are some anomalous results. For example, in AndroidWorld under the Mixture‑model setting, applying GUI‑PRA (W/ GUI‑PRA‑Q) shows no improvement over PRM (W/ PRM‑Q); in Mobile‑MiniWoB++, the PRM method (W/ PRM‑I) even performs slightly worse than the baseline. How do the authors explain these inconsistencies?
- Line 397 states that “GUI‑PRA’s most significant advantages emerge on tasks of medium difficulty.” Why does the proposed approach yield substantial improvement only at the medium‑difficulty level? What are the essential differences among the difficulty categories?
- The ablation study is conducted only on the Qwen‑VL series. Can the same component‑wise conclusions be generalized to the InternVL or Mixture‑model settings?

---

### Official Review · Reviewer_jDFA · 2025-11-01

**Soundness:** 1
**Presentation:** 2
**Contribution:** 2
**Rating:** 2
**Confidence:** 3

**Summary:**

This paper address a key limitation identified in prior work, that is long-horizon reasoning for GUI agents due to context length constraints. The topic is timely and of clear interest to the community, as there are many different cuncurrent work. The authors ground their formulation in the “Lost in the Middle”, and propose GUI-PRA, which extends an existing framework Process Reward Agents (PRAs) for GUI tasks through two main components:

1) Summarizes earlier interaction history within a ReAct-style chain and appends it to later steps as a compact context signal for the reward critique model.
2) Enables the model to choose between two perception tools: one for coarse and one for fine-grained visual understanding during evaluation.

**Strengths:**

1) The paper tackles a well-motivated problem: the long-horizon performance degradation of GUI agents.

2) The discussion of “lost in the middle” and dynamic memory aligns well with known LLM context limitations, making the contribution conceptually coherent.

3) The approach is conceptually clean, adapting the PRM framework to GUI-specific challenges of long context and visual grounding.

4) The method is evaluated on two online benchmarks (AndroidWorld and MobileMiniWoB++)

**Weaknesses:**

1.	Limited Novelty in Core Ideas
The proposed summarization module overlaps substantially with mechanisms already introduced in recent GUI-agent literature. For example, GUI-Rise: Structured Reasoning and History Summarization for GUI Navigation (Liu et al.) also introduces a dedicated “history summarization” subtask. Thus, the novelty lies more in applying such summarization within a PRM context rather than in the idea itself. Outside of GUI there are lot of papers that do the sumarization to handle long context. Summarise earlier interaction chains (ReAct style) to fetch relevant history on PRA for GUI is very incremental in terms of novelty.

2. Allowing the model to select between “coarse” and “fine” visual tools resembles prior multi-granularity perception designs (e.g., Read Anywhere Pointed, Attention-Driven GUI Grounding). The contribution appears to be a selection mechanism over pre-existing tools, rather than a fundamentally new perception module. Again this means the conceptual jump is small. Unless I missed a key point, I think the proposed pipeline overall lacks novelty.

3. The benchmarking is limited to two datasets (AndroidWorld and MobileMiniWoB++). Several popular and relevant long-horizon benchmarks—such as WebArena, Mind2Web, OS-World, and WorkArena—are omitted, despite being directly adaptable to the proposed setting.
The ablations are also not exhaustive, and many strong baselines and concurrent works are missing from comparison (e.g., Aria-UI, Autonomous Evaluation and Refinement of Digital Agents, GUI-LLM). Even excluding the arxiv ones, there is prior work like 'Autonomous Evaluation and Refinement of Digital Agents' to be adapted to and compared with proposed method. They study different evaluators and improve the performance of existing agents via fine-tuning and inference-time guidance.
Not a reason for lower rating but the quantitative improvements reported (e.g., Table 2) are relatively small and not always consistent across datasets or settings. The results appear as early evidence rather than conclusive validation.


4. The related work section omits several closely related studies, including Lost in the Middle itself, which is central to the paper’s formulation. Broader citation coverage of process supervision and GUI perception literature would strengthen the work’s positioning: "Lost in the Middle: How Language Models Use Long Contexts".

**Questions:**

1.	Can you include more benchmarks such as Mind2Web and WebArena, which explicitly involve long-horizon web or GUI tasks?
2.	Can the authors report the distribution of task lengths (e.g., number of actions per episode) and how the proposed method performs as task length increases?
3.	A plot of performance gain versus task length would clarify the practical utility of the proposed dynamic memory mechanism.
4.	Does it make sense to include an ablation for each part with related work like discussing GUI-LLM (CVPR 2024), wfor multimodal perception for GUI reasoning.

---

### Official Review · Reviewer_M8uW · 2025-11-02

**Soundness:** 3
**Presentation:** 3
**Contribution:** 3
**Rating:** 2
**Confidence:** 4

**Summary:**

The paper introduces a training-free process reward adjudicator (GUI-PRA) for mobile GUI agents. At each step it builds a compact memory of recent interactions and queries on-screen perception tools, then re-scores multiple candidate thought-action options and executes the best one. On AndroidWorld and Mobile-MiniWoB++, this test-time controller boosts end-to-end task success over both vanilla agents and a standard PRM baseline.

**Strengths:**

Practical and model-agnostic (drop-in, no finetuning). Combines dynamic memory with explicit UI perception to curb long-horizon drift and brittle text-only judging. Shows consistent gains across two benchmarks and different backbones, with ablations that clarify where improvements come from. Orthogonal to underlying planners/PRMs, so it can stack with stronger agents and search methods.

**Weaknesses:**

In ablations, removing Point (local UI targeting) drops performance to below standard PRM (−0.44%), indicating the overall gains hinge on the reliability of OmniParser/Point and the routing policy’s correctness.


Both standard PRM and GUI-PRA do an argmax over k candidate thought–action pairs per step. If the base agent doesn’t propose the correct action, re-scoring cannot recover it.


Error propagation via “temporal consistency.” The scorer feeds the previously selected action and its score into the current step to enforce consistency. An early misjudgment can bias the remainder of the trajectory.


The paper claims to assess both “effectiveness and efficiency,” but only reports SR/DSR, no latency, token usage, or tool-call counts, making cost–benefit hard to judge.

Results are limited to mobile (AndroidWorld and Mobile-MiniWoB++). Desktop OS scenarios, multi-app workflows, cross-app long-horizon tasks, dynamic window management, aren’t covered, so generalization to OSWorld/desktop remains unproven.

**Questions:**

See Above

---

### Official Review · Reviewer_qcKY · 2025-11-03

**Soundness:** 3
**Presentation:** 3
**Contribution:** 2
**Rating:** 4
**Confidence:** 2

**Summary:**

This paper introduces GUI-PRA (Process Reward Agent for GUI Tasks), a novel training-free framework designed to enhance the performance of Process Reward Models (PRMs) in graphical user interface (GUI) automation. The paper highlights that standard PRMs suffer from a “lost in the middle” phenomenon when handling long task histories and lack awareness of UI state changes.

To address these limitations, GUI-PRA incorporates a Dynamic Memory Mechanism to condense historical trajectories and an Adaptive UI Perception Mechanism to actively gather visual evidence, thereby providing more accurate supervisory signals. Experimental results demonstrate that GUI-PRA achieves substantially higher success rates than both the baseline and standard PRM-guided models across two online benchmarks, validating its effectiveness in improving the reliability and efficiency of complex GUI task execution.

**Strengths:**

(1) Introduces a new training-free supervisory agent paradigm that can generalize to unseen GUI tasks.

(2) Empirical improvement (~14.5%) is meaningful for GUI automation, where success rates are typically low.

**Weaknesses:**

(1)	 Limitations of Cross-Family Generalization and Lack of Explanation. Although the experimental setup includes an evaluation of cross-family generalization—where the InternVL3 Agent is guided by the Qwen2.5-VL Supervisor—the results indicate that GUI-PRA fails to deliver any additional improvement over the standard PRM under this mixed-model configuration. However, the authors provide no explanation for this outcome. Moreover, this finding appears to contradict the paper’s first claim in the introduction that “GUI-PRA, a novel agent that surpasses standard PRMs for GUI tasks.”

(2)	One of the core contributions of the paper is the Dynamic Memory Mechanism. However, its implementation details raise concerns about rigor and transparency. The definition of “relevance” is unclear: the paper describes the retrieval stage as a “Relevance-based Retrieval” process intended to actively fetch pertinent information. Yet, in the formal description, the retrieval function 𝑓 retrieve f retrieve is simply defined as retaining the most recent m steps. How does this ensure that the selected window of m steps is truly relevant to the task goal?

**Questions:**

Please see the weaknesses section.

---

### Note · Authors · 2026-01-06

**Comment:**

Dear Area Chairs and Reviewers,

After carefully reading the reviews and considering the feedback provided, we have decided to withdraw our submission.

We would like to express our sincere gratitude to the reviewers for their time and effort in evaluating our work. The constructive comments and detailed suggestions are very valuable to us. We will use this feedback to improve our manuscript for a future submission.

Thank you again for your hard work.

Best regards

**Withdrawal Confirmation:**

I have read and agree with the venue's withdrawal policy on behalf of myself and my co-authors.